# Electrically Controlled Diffraction Grating in Azo Dye-Doped Liquid Crystals

**DOI:** 10.3390/polym11061051

**Published:** 2019-06-16

**Authors:** Chuen-Lin Tien, Rong-Ji Lin, Chi-Chung Kang, Bing-Yau Huang, Chie-Tong Kuo, Shuan-Yu Huang

**Affiliations:** 1Department of Electrical Engineering, Feng Chia University, Taichung 40724, Taiwan; cltien@fcu.edu.tw; 2Ph.D. Program of Electrical and Communications Engineering, Feng Chia University, Taichung 40724, Taiwan.; jimmylin2002x@gmail.com; 3Department of Optometry, Da-Yeh University, Changhua 515, Taiwan; 4Department of Physics and Center for Nanoscience and Nanotechnology, National Sun Yat-San University, Kaohsiung 804, Taiwan; Kangchi@ymail.com (C.-C.K.); flyfish31@hotmail.com (B.-Y.H.); 5Department of Optometry, Chung Shan Medical University, Taichung 402, Taiwan; 6Department of Ophthalmology, Chung Shan Medical University Hospital, Taichung 402, Taiwan

**Keywords:** beam spiltting, grating-like electrodes, birefringence, nematic liquid crystals, azo dye

## Abstract

This research applies the non-linear effect of azo dye-doped liquid crystal materials to develop a small, simple, and adjustable beam-splitting component with grating-like electrodes. Due to the dielectric anisotropy and optical birefringence of nematic liquid crystals, the director of the liquid crystal molecules can be reoriented by applying external electric fields, causing a periodic distribution of refractive indices and resulting in a diffraction phenomenon when a linearly polarized light is introduced. The study also discusses the difference in the refractive index (*Δn*), the concentration of azo dye, and the rising constant depending on the diffraction signals. The experimental results show that first-order diffraction efficiency can reach ~18% with 0.5 wt % azo dye (DR-1) doped in the nematic liquid crystals.

## 1. Introduction

Diffraction grating is an optical element with periodic distribution of thicknesses or refractive indices, which can decompose and diffract optical waves. The diffracted direction depends on the periodic of thickness or refractive index of diffraction grating and the wavelength of the incident light [1]. With this feature, diffraction grating is an important component in optical spectroscopy [2]. In addition, tunable gratings have high potential on displays, sensors, and tunable lasers [3,4,5,6,7,8]. Several methods and materials have been employed to fabricate tunable gratings, such as thermal actuation [9], electrostatic actuation [10], microfluidic actuation [11], piezoelectric materials [12], elastomers [13,14,15], and liquid crystals (LCs) [16,17,18,19,20,21]. 

Controllable gratings based on liquid crystals have attracted significant attention, because of several superior advantages such as ease of fabrication and operation. Azo dye-doped liquid crystals and related materials are employed for controllable holographic gratings [22,23,24]. In addition, the photo-induced isomerization of azo dyes [25] and thus the photo-induced reorientations [26], photo thermal effect [27], or photo-induced isothermal phase transitions [28] in azo dye-doped liquid crystals are widely studied and applied on non-linear optics, photo alignment, and photo actuators [29,30,31]. In this work, we present a prototype of electrically controllable diffraction gratings based on liquid crystals. By applying an electric field on the grating-like electrodes, diffraction efficiency can be tuned. The diffraction efficiency of the tunable grating device can be further improved by doping a tiny amount of azo dye. However, when the concentration of dye exceeds a critical value, the diffraction efficiency decreases due to the excess dye molecules and thus the disturbance of liquid crystal orientations.

## 2. Materials and Methods

The diffraction gratings demonstrated in this work were fabricated by injecting azo dye-doped liquid crystals into empty cells with grating-like electrodes. The azo dye-doped liquid crystal was composed of nematic liquid crystal (E7, from Merck, Darmstadt, Germany) and azo dye (DR-1, from TCI, Tokyo, Japan). To investigate the effect of dye concentration on the performance of the grating, five concentrations of azo dye-doped LCs mixtures were prepared. The concentrations of the azo dye were 0, 0.25, 0.50, 1.0, and 1.5 wt %, respectively. The azo dyes in powder form were added into the nematic liquid crystals directly and then sonicated to obtain the homogeneous mixtures. To fabricate the empty cell, one glass substrate with grating-like electrodes and one ITO-coated glass substrate were assembled with two 38-μm-thick Mylar spacers. A substrate with grating-like electrodes was fabricated by photolithography. First, the positive photoresist AZ1500 was spin-coated on an ITO-coated glass substrate at 7000 rpm for 30 s and then soft baked at 90 °C for 30 s. Next, a photomask with a grating-like pattern was aligned with the photoresist-coated substrate and exposed under an exposure system for 10 s. After exposure, the substrate was baked at 90 °C for 30 s. The substrate was immersed in the developer AZ400K for 13 s and then washed by deionized (DI) water to form the patterned photoresist. The substrate was baked at 90 °C for 30 s to harden the photoresist. The substrate was then immersed into a HCl solution (37 wt %) for 35 s to etch the ITO of the exposed areas. Finally, the residue photoresist was then removed by acetone and the process for fabricating the grating-like electrodes was finished. The width and space of the grating-like electrode were 20 and 40 μm, respectively. To provide a pre-orientated direction for the liquid crystal molecules, the substrates were coated with alignment layers (SE-130, from Nissan, Tokyo, Japan) and anti-parallel rubbed.

A real-time probe technique was employed to detect the dynamic process of the diffraction grating. As shown in Figure 1, a He-Ne laser with a pair of polarizers and a half-wave plate was aligned as the probe beam. The diameter of the probe beam was about 3 mm. Arbitrary intensity and arbitrary linear polarization of the probe beam can be obtained by properly adjusting the half-wave plate and polarizer. An alternating current (AC) field with frequency of 1 kHz was applied on the sample to induce the reorientations of liquid crystal molecules and thus to form the diffraction grating. Two photo detectors (ET-2040, from EOT, Traverse, MI, US) were set behind the sample to receive the zero-order and first-order signals, respectively. It should be noted that the orientation for the period of the grating electrode was set in the x-axis, while the initial orientation of liquid crystal molecules was along the y-axis, as seen in the coordinate provided in Figure 1. The polarization of the probe beam was set to be parallel to the y-axis.

## 3. Results and Discussion

When the applied voltage is 0 V, the director of all the liquid crystal molecules in the sample is parallel to the y-axis. No diffraction occurs due to the homogeneous distribution of the refractive index at 0 V. The distribution of the refractive index can be induced periodically by an electric field. Figure 2 presents the diffraction phenomena when the sample is operated with various applied voltages. When the applied voltage is around 1.2 V, which is the threshold voltage (*V*_th_), the liquid crystal molecules in the electrode region can be oriented slightly by the electric field, and thus the diffraction phenomenon is still not obvious. As the voltage increases to 5 V, the liquid crystal molecules in the electrode region will almost orient themselves to be parallel to the z-axis, while the liquid crystal molecules in the non-electrode region will remain in the direction of the y-axis. Since the polarization of the incident light is y-polarized, the difference in the refractive index experienced by the incident light between the electrode and non-electrode regions is significant. In other words, the detected beam experiences the maximum difference of refractive indices (*Δn*) and results in the highest first-order diffraction when the applied voltage is 5 V. As the voltage increases to 8.5 V, the electric field is large enough to reorient the liquid crystal molecules even in the non-electrode strips, and, consequently, the *Δn* experienced by the detected beam becomes smaller, resulting in lower diffraction intensity. When the polarization of the incident probe beam rotates from the y-axis to the x-axis, the refractive index experienced by the beam in the non-electrode regions is almost the same as that in the electrode regions remains. The reduced variation of refractive indices between the electrode and non-electrode regions weakens the effect of the grating. Therefore, the diffraction efficiency decreases when the polarization of the incident probe beam deviates from the y-axis. Similar results can be found in a previous study [32].

We now define the diffraction efficiency as the ratio of the first-order diffraction intensity to the incident beam intensity, as represented by Equation (1):(1)$$\eta =\frac{I_{1}}{I_{0}}$$
where *I*_0_, and *I*_1_ are the intensities of the incident beam and the first-order diffraction, respectively. According to the Kogelnik formula, the diffraction efficiency can be calculated by Equation (2) [33]:(2)$$\eta =\sin^{2}\frac{\pi \Delta nd}{\lambda \cos\beta }$$
where *λ* and *β* are the respective wavelength and the incident angle of the probe beam, *d* is the thickness of the liquid crystal layer, and *Δn* is the difference of the refractive indices in the grating. For the case of the sample operated at 5 V, the diffraction efficiency *η* is 12.5%. If we substitute *η* = 12.5%, *d* = 38 μm, *λ* = 0.633 μm, and *β* = 0° into Equation (2), then we estimate that *Δn* = 0.0019.

Figure 3a shows the first-order diffraction as a function of time during the period between 0 to 1000 ms with an applied voltage of 5.0V. The diffraction signal rapidly rises to a stable level. The rising curve of the signal versus time can be fitted by a single exponential function as in Equation (3):(3)$$I(t)=I_{s}(1-e^{\frac{-t}{\tau }})$$
where *I*_s_ is the saturated diffraction intensity, and *τ* is the rise constant of the response time obtained from the simulation of the rising curve. Here, *I*_s_ and *τ* in the fitting curve are 0.068 and 128.7 ms, respectively. From Equation (3), the rise time of the diffraction (response time for the intensity of 10% of the maximum to 90% of the maximum) operated at 5 V is estimated as 283 ms. When the voltage is switched off, the diffraction intensity falls to almost zero, as shown in Figure 3b. The fall time of the diffraction is about 300 ms. During the falling process, a transient rise of the diffraction intensity can be observed. This phenomenon is believed to result from perturbation and reorientation of the liquid crystal director after the voltage is removed. The experimental results indicate that the electrically controlled diffraction in the liquid crystal grating is reversible.

Figure 4 shows the maximum value of the first-order diffraction with varying weight concentration of the dye in the liquid crystals. As shown in Figure 4, the first-order diffraction efficiency increases at first and then decreases with an increase of the concentration of dye. The initial rise of the diffraction signal with an increasing concentration of dye is due to fact that the azo dye molecules are rearranged by liquid crystal molecules caused by the guest–host effect [34]. At this stage, the probe beam experiences not only the difference of the refractive indices from the liquid crystal molecules, but also the difference of the refractive indices from the dye molecules. A superposition of the difference of the refractive indices from the liquid crystal molecules and the dye molecules results in an increase of diffraction efficiency. When the doped concentration of the azo dye is 0.5 wt %, the diffraction efficiency improves by about 45%, compared to the pure liquid crystal sample. However, when the dye concentration increases to some extent, the dye molecules disturb the order of the liquid crystals rearranged under the electric field. Consequently, the difference in the refractive index perceived by the detection light does not increase and thus results in a decrease in diffraction efficiency.

The effective differences of the refractive indices in the liquid crystal grating devices with various concentrations of azo dyes can be estimated by substituting the diffraction efficiencies shown in Figure 4 into Equation (2). The results are listed in Table 1. One can find that the effective difference of the refractive indices increases and then decreases with a rise in the dye concentration. This result corresponds to our previous statement—that is, the order parameter of liquid crystal molecules and thus the effective difference of refractive indices can be improved by doping a tiny amount of dye. When the concentration of dye exceeds a critical value, the existence of dyes does not benefit the orientation of liquid crystal molecules and, thus, reduces diffraction efficiency.

## 4. Conclusions

This paper has demonstrated electrically controllable diffraction gratings by combining liquid crystals and designed electrodes. By applying an electric field, the orientation of liquid crystal molecules can be controlled and the diffraction efficiency can be tuned. When the applied voltage is 5 V, the device exhibits the best diffraction efficiency. We have also found that the diffraction efficiency of the tunable grating device can be improved by about 45% by doping 0.5 wt % of azo dye. The additional difference of refractive indices from the dye molecules is believed to be the main factor for the improved performance. When the dye concentration is high, the excess amount of dye disturbs the orientation of liquid crystal molecules and thus decreases diffraction efficiency. The results of the doping effect reported herein provide a new insight into the development of liquid crystal devices.

## Figures and Tables

**Figure 1 polymers-11-01051-f001:**
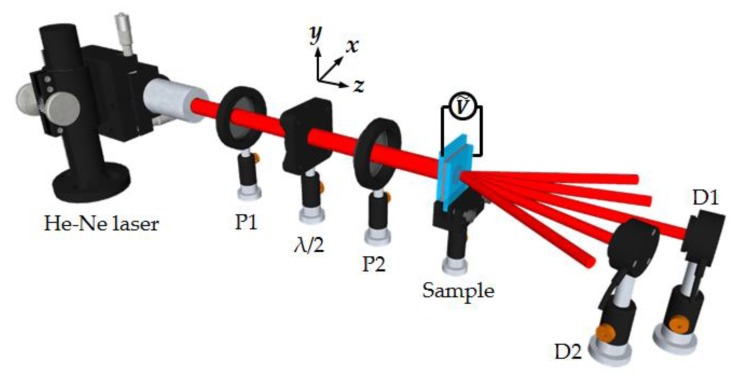
Schematic illustration of the experimental set-up. P1, P2: polarizers; λ/2: half-wave plate; D1, D2: detectors.

**Figure 2 polymers-11-01051-f002:**
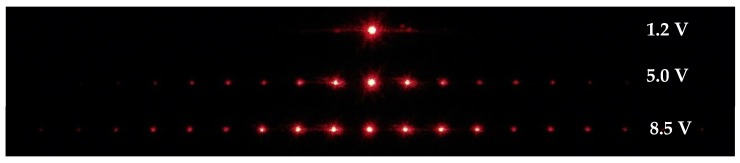
Images of the diffraction when the device is operated with the applied voltages of 1.2, 5.0, and 8.5 V, respectively.

**Figure 3 polymers-11-01051-f003:**
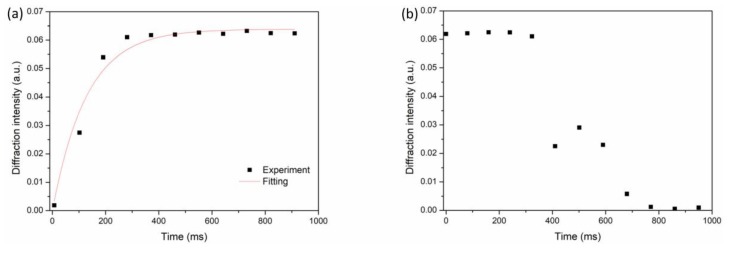
Diffraction efficiency of the first-order diffraction as a function of time (**a**) when the device is operated with an applied voltage of 5.0 V, and (**b**) when the applied voltage is switched off.

**Figure 4 polymers-11-01051-f004:**
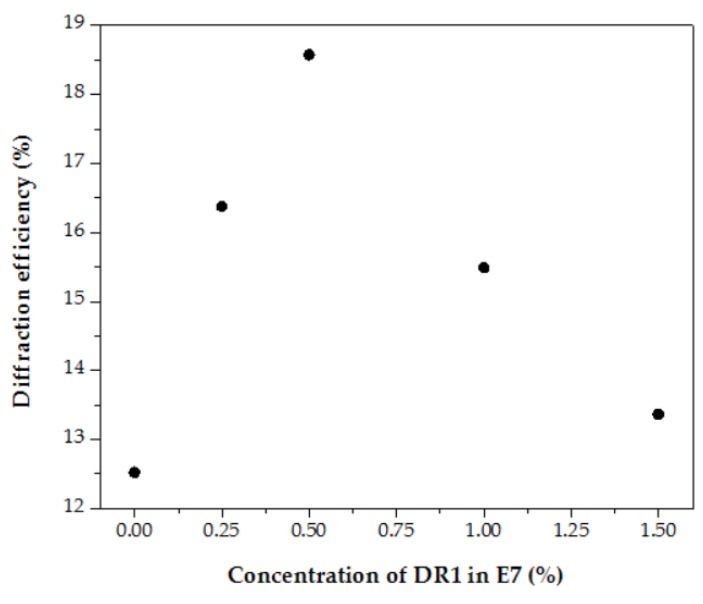
Diffraction efficiency of the first-order diffraction as a function of the concentration of azo dye. The devices are operated under an applied voltage of 5.0 V.

**Table 1 polymers-11-01051-t001:** Diffraction efficiencies and the corresponding differences of refractive indices in samples with various dye concentrations.

Concentration of Dye (wt %)	Diffraction Efficiency (%)	*Δn*
0	12.5	0.0020
0.25	16.2	0.0022
0.50	18.2	0.0023
1.00	15.6	0.0022
1.50	13.4	0.0020

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
