# Peer review of "Electrically Controlled Diffraction Grating in Azo Dye-Doped Liquid Crystals"

_polymers, 2019, doi:10.3390/polym11061051_

Round 1
Reviewer 1 Report
In this paper, the authors report the increase of diffraction efficiencies by doping of azo dye in a liquid crystal material and application of this system in developing a tunable beam splitting component with grating-like electrodes.
Tunable gratings have many advantages which provoke significant attention among researchers. The reported results are relevant and important.
Here are several remarks (not listed in order of importance), which have to be considered:
1. Some points from the materials and methods section should be explained more explicitly. Such as fabrication of the empty cell, with one glass substrate with grating-like electrodes. How was this substrate with electrodes prepared? What material are the electrodes made from? Line 51. Another one: nothing is explained about the intensity of the probe beam and beam spot size. Also, there is no information or discussion for the influence of the effect from polarization.
2. When we define the diffraction efficiency we usually use the ratio of the first order diffraction intensity to the incident beam intensity. If the authors want to use some other ratio, they must change the term used.
3. To be precise in Fig.1, the polarization after the second linear polarizer can be only "arbitrary linear", but not "arbitrary". Line 58.
4. One technical remark. In line 26 the first keyword is “beam spilt” instead of” beam split”.
5. It may have some misunderstanding with the orientation of the different elements of the cell. By definition the grating vector has orientation perpendicular to the grating plane, but the authors state that it is in x direction, which is in the glass plane and it is not perpendicular to the grating plane at the fig. 1. It would be best that the authors represent schematically the cell and orientation of the grating electrodes and LC molecules or to clarify the text.
6. Here in this paper the authors just declared that highest first order diffraction is at 5V without showing the results. This study is presented in one of their previous works when they used only LC without doped dye. That is why at least citation of this work from which these results are assumed should be included
7. Highest concentration of dye still shows higher DE from pure LC. Have the authors tried with higher value that shows decrease from 0 wt% concentration?
8. In the introduction section other scientific achievements with doping of azo dye in liquid crystal material should be included. Azo materials have been studied for more than 30 years and there are many scientific groups working in this area.
Author Response
1. Some points from the materials and methods section should be explained more explicitly. Such as fabrication of the empty cell, with one glass substrate with grating-like electrodes. How was this substrate with electrodes prepared? What material are the electrodes made from? Line 51. Another one: nothing is explained about the intensity of the probe beam and beam spot size. Also, there is no information or discussion for the influence of the effect from polarization.
Response:
To answer the reviewer’s questions, we added some sentences in Materials and Methods to describe the fabrication process of the grating-like electrodes:
“A substrate with grating-like electrodes was fabricated by photolithography. First, the positive photoresist AZ1500 was spin coated on an ITO-coated glass substrate at 7000 rpm for 30 s and then soft baked at 90 ºC for 30 s. Next, a photomask with a grating-like pattern was aligned with the photoresist-coated substrate and exposed under an exposure system for 10 s. After exposure, the substrate was baked at 90 ºC for 30 s. The substrate was immersed in the developer AZ400K for 13 s and then washed by DI water to form the patterned photoresist. The substrate was baked at 90 ºC for 30 s to harden the photoresist. The substrate was then immersed into a HCl solution (37 wt%) for 35 s to etch the ITO of the exposed areas. Finally, the residue photoresist was then removed by acetone and the process for fabricating the grating-like electrodes was finished.” at lines 58-67 of the revised manuscript.
A sentence “The diameter of the probe beam was about 3 mm.” has been inserted into line 72 of the revised manuscript to supplement the information of the probe beam.
To discuss the influence of the effect from polarization, we have added a sentence “When the polarization of the incident probe beam rotates from y-axis to x-axis, the refractive index experienced by the beam in the non-electrode regions is almost the same as that in the electrode regions remains. The reduced variation of refractive indices between the electrode and non-electrode regions weakens the effect of the grating. Therefore, the diffraction efficiency decreases when the polarization of the incident probe beam deviates from the y-axis.” at lines 102-106 of the revised manuscript.
2. When we define the diffraction efficiency we usually use the ratio of the first order diffraction intensity to the incident beam intensity. If the authors want to use some other ratio, they must change the term used.
Response:
We apologize for the wrong definition in the origin manuscript. After checking the raw data carefully, we confirm that the diffraction efficiency we measure is the ratio of the first order diffraction intensity to the incident beam intensity, as described by the reviewer. Therefore, we correct our definition of diffraction efficiency in the revised manuscript. The term “the zeroth order diffraction intensity” has been changed as “the incident beam intensity” at line 114 of the revised manuscript; “the intensities of the zeroth and the first order diffraction” has been changed as “the intensities of the incident beam and the first-order diffraction” at line 116 of the revised manuscript.
3. To be precise in Fig.1, the polarization after the second linear polarizer can be only "arbitrary linear", but not "arbitrary". Line 58.
Response:
We really appreciate the reviewer for the correction. The word “polarization” has been modified as “arbitrary linear polarization” at line 73 of the revised manuscript.
4. One technical remark. In line 26 the first keyword is “beam splitting” instead of “beam split”.
Response:
The first keyword has been corrected as “beam splitting” as suggested by the reviewer.
5. It may have some misunderstanding with the orientation of the different elements of the cell. By definition the grating vector has orientation perpendicular to the grating plane, but the authors state that it is in x direction, which is in the glass plane and it is not perpendicular to the grating plane at the fig. 1. It would be best that the authors represent schematically the cell and orientation of the grating electrodes and LC molecules or to clarify the text.
Response:
We apologize that the word grating vector was used improperly. To avoid misunderstanding, the term “the direction of the grating vector” has been modified to “the orientation for the period of the grating electrode” at lines 77-78 of the revised manuscript.
6. Here in this paper the authors just declared that highest first order diffraction is at 5V without showing the results. This study is presented in one of their previous works when they used only LC without doped dye. That is why at least citation of this work from which these results are assumed should be included
Response:
A sentence “. Similar results can be found in a previous study [32].” has been inserted at lines 106-107 of the revised manuscript and a new reference has also been cited.
Newly cited reference:
32. Tien, C.-L.; Lin, R.-J.; Su, S.-H.; Horng, C.-T. Electrically tunable diffraction grating based on liquid crystals. Adv. Cond. Matter Phys. 2018, 2018, 7849529.
7. Highest concentration of dye still shows higher DE from pure LC. Have the authors tried with higher value that shows decrease from 0 wt% concentration?
Response:
Due to the limit of solubility, the highest concentration of azo dye we have tried is 1.5 wt% Because excess amount of dyes will aggregate, we believe the diffraction efficiency will decrease if the concentration of the azo dye increases further.
8. In the introduction section other scientific achievements with doping of azo dye in liquid crystal material should be included. Azo materials have been studied for more than 30 years and there are many scientific groups working in this area.
Response:
As suggested by the reviewer, we have added a statement to briefly introduce azo dye in liquid crystal materials at lines 38-43 of the revised manuscript: “Azo dye-doped liquid crystals and related materials are employed for controllable holographic gratings [22−24]. In addition, the photo induced isomerization of azo dyes [25] and thus the photo induced reorientations [26], photo thermal effect [27], or photo induced isothermal phase transitions [28] in azo dye-doped liquid crystals are widely studied and applied on non-linear optics, photo alignment, and photo actuators [29−31].”
Newly cited references:
22. Lee, C.-R.; Mo, T.-S.; Cheng, K.-T.; Fu, T.-L.; Fu, A.Y.-G. Electrically switchable and thermally erasable biphotonic holographic gratings in dye-doped liquid crystal films. Appl. Phys. Lett. 2003, 83, 4285–4287.
23. Presnyakov, V.; Asatryan, K.; Galstian, T.; Chigrinov, V. Optical polarization grating induced liquid crystal micro-structure using azo-dye command layer. Opt. Express 2006, 14, 10558–10564.
24. Ho, T.J.; Chen, C.W.; Khoo, I.C. Polarisation-free and high-resolution holographic grating recording and optical phase conjugation with azo-dye doped blue-phase liquid crystals. Liq. Cryst. 2018, 45, 13–15.
25. Bandara, H.M.D.; Burdette, S.C. Photoisomerization in different classes of azobenzene. Chem. Soc. Rev. 2012, 41, 1809–1825.
26. Jánossy, I.; Szabados, L. Optical reorientation of nematic liquid crystals in the presence of photoisomerization. Phys. Rev. E 1998, 58, 4598–4604.
27. Lee, K.M.; White, T.J. Photochemical mechanism and photothermal considerations in the mechanical response of monodomain, azobenzene-functionalized liquid crystal polymer networks. Macromolecules 2012, 45, 7163–7170.
28. Ikeda, T.; Horiuchi, S.; Karanijit, D.B.; Kurihara, S.; Tazuke, S. Photochemically induced isothermal phase transition in polymer liquid crystals with mesogenic phenyl benzoate side chains. 1. Calorimetric studies and order parameters. Macromolecules 1990, 23, 36–42.
29. Khoo, I.C.; Li, H.; Liang, Y. Optically induced extraordinarily large negative orientational nonlinearity in dye-doped liquid crystal. IEEE J. Quantum Electron. 1993, 29, 1444–1447.
30. Chigrinov, V; Muravski, A.; Kwok, H.S. Anchoring properties of photoaligned azo-dye materials. Phys Rev. E 2003, 68, 061702.
31. Jiang, H.; Li, C.; Hunag, X. Actuators based on liquid crystalline elastomer materials. Nanoscale 2013, 5, 5225–5240.

Reviewer 2 Report
The manuscript presents electrically controllable diffraction gratings based on liquid crystal doped with the azo dye DR-1. An optimal concentration of the azo dye is reported (0.5 wt. %) that leads to 45% increase of the diffraction efficiency of the grating.
The study is interesting and can be useful for researchers in the field of diffractive optics and liquid crystal devices. Still, some important questions and issues must be addressed by the authors. My comments are listed according to the line numbers in the manuscript as follows:
Line 2-3: A revision of the paper title is recommendable, as it does not reflect the specifics of the research presented. I believe that “Electrically Controlled Diffraction Grating in Azo Dye doped Liquid Crystals” reflects better the paper’s topic.
Line 26: “beam spilt” should become “beam splitting”
Lines 39, 41, 138, 142: The term “tunable diffraction grating” usually refers to the possibility to vary the grating period, not its diffraction efficiency. Therefore, I strongly recommend to the authors to use the term “electrically controllable” instead of “electrically tunable” everywhere in the article.
Line 50: More details must be given about the procedure to prepare the LC – azo dye mixture. Is the azo dye initially dissolved in some solvent or it is added to the LC in powder form? Is the mixture then sonicated or treated in some way?
Line 59: Is the electric field applied AC or DC? If it is indeed AC, what is the frequency used?
Line 68: More data should be presented about the detectors D1 and D2, at least model and measurement uncertainty.
Lines 74 and 85, 86: In the text, the value of the applied voltage is 1.5 V, but in Figure 2 and its caption the value is 1.2 V. Which is the correct one or are the values intentionally different? This must be clarified.
Lines 88 – 91: The diffraction efficiency is defined as the ratio between the diffracted intensity and the incident intensity, not the zeroth order intensity! For example, if the intensity diffracted in the first order is 75% and in the zeroth order 25% of the incident, using Eq. 1 (line 90) we will obtain 300% diffraction efficiency, which is nonsense. Hence, all the diffraction efficiencies in the paper must be recalculated, in order for the results to be comparable with other research papers.
Line 104: A dimension is missing. Should be “128.7 ms”.
Lines 97 – 108: What is the time constant when the applied voltage is switched off? Is the process fully reversible i.e. does the diffraction efficiency fall to zero, or there is some residual diffraction? I suggest to modify Figure 3 and add at least 2-3 cycles of switching the voltage on and off.
Line 136: After recalculating the diffraction efficiencies, the modulation of the refractive indices (Δn) should also be recalculated. Furthermore, the values of Δn should be given with not more than 2 significant figures (e.g. 0.0019 instead of 0.001917). Giving values with 4 significant figures requires much higher precision of measuring the diffraction efficiency and LC cell thickness.
Lines 148 – 155: Author contributions and Funding sections are not filled in completely.
Author Response
The manuscript presents electrically controllable diffraction gratings based on liquid crystal doped with the azo dye DR-1. An optimal concentration of the azo dye is reported (0.5 wt. %) that leads to 45% increase of the diffraction efficiency of the grating.
The study is interesting and can be useful for researchers in the field of diffractive optics and liquid crystal devices. Still, some important questions and issues must be addressed by the authors. My comments are listed according to the line numbers in the manuscript as follows:
Line 2-3: A revision of the paper title is recommendable, as it does not reflect the specifics of the research presented. I believe that “Electrically Controlled Diffraction Grating in Azo Dye doped Liquid Crystals” reflects better the paper’s topic.
Response:
We are grateful for the valuable suggestion. As suggested, the title of the revised manuscript has been modified as “Electrically Controlled Diffraction Grating in Azo Dye Doped Liquid Crystals”.
Line 26: “beam spilt” should become “beam splitting”
Response:
The word “beam split” has been modified as “beam splitting”.
Lines 39, 41, 138, 142: The term “tunable diffraction grating” usually refers to the possibility to vary the grating period, not its diffraction efficiency. Therefore, I strongly recommend to the authors to use the term “electrically controllable”instead of “electrically tunable” everywhere in the article.
Response:
The terms “electrically tunable” have been modified as “electrically controllable” as suggested by the reviewer.
Line 50: More details must be given about the procedure to prepare the LC – azo dye mixture. Is the azo dye initially dissolved in some solvent or it is added to the LC in powder form? Is the mixture then sonicated or treated in some way?
Response:
To response to the reviewer’s question, we have inserted a sentence “The azo dyes in powder form were added into the nematic liquid crystals directly and then sonicated to obtain the homogeneous mixtures.” at lines 55-56 of the revised manuscript.
Line 59: Is the electric field applied AC or DC? If it is indeed AC, what is the frequency used?
Response:
The term “AC field with frequency of 1kHz” has been inserted at line 74 of the manuscript.
Line 68: More data should be presented about the detectors D1 and D2, at least model and measurement uncertainty.
Response:
The model of the detectors and their manufacturer has been added as “(ET-2040, from EOT)” at line 76
Lines 74 and 85, 86: In the text, the value of the applied voltage is 1.5 V, but in Figure 2 and its caption the value is 1.2 V. Which is the correct one or are the values intentionally different? This must be clarified.
Response:
We apologize for our ignorance. The value of the applied voltage at line 91 has been corrected from 1.5 V to 1.2 V.
Lines 88 – 91: The diffraction efficiency is defined as the ratio between the diffracted intensity and the incident intensity, not the zeroth order intensity! For example, if the intensity diffracted in the first order is 75% and in the zeroth order 25% of the incident, using Eq. 1 (line 90) we will obtain 300% diffraction efficiency, which is nonsense. Hence, all the diffraction efficiencies in the paper must be recalculated, in order for the results to be comparable with other research papers.
Response:
We apologize for the wrong definition in the origin manuscript. After checking the raw data carefully, we confirm that the diffraction efficiency we measure is the ratio of the first order diffraction intensity to the incident beam intensity, as described by the reviewer. Therefore, we correct our definition of diffraction efficiency in the revised manuscript. The term “the zeroth order diffraction intensity” has been changed as “the incident beam intensity” at line 114 of the revised manuscript; “the intensities of the zeroth and the first order diffraction” has been changed as “the intensities of the incident beam and the first order diffraction” at line 116 of the revised manuscript.
Line 104: A dimension is missing. Should be “128.7 ms”.
Response:
We thank the reviewer for the careful review. The dimension “ms” has been added at line 129 of the revised manuscript.
Lines 97 – 108: What is the time constant when the applied voltage is switched off? Is the process fully reversible i.e. does the diffraction efficiency fall to zero, or there is some residual diffraction? I suggest to modify Figure 3 and add at least 2-3 cycles of switching the voltage on and off.
Response:
To address on the reviewer’s comment, we have added Fig. 3(b) in the revised manuscript to show the falling process of the diffraction intensity after switching off the applied voltage. The original Fig. 3 has also been rearranged as Fig. 3(a). In addition, a statement When the voltage is switched off, the diffraction intensity falls to almost zero, as shown in Fig. 3(b). The fall time of the diffraction is around 300 ms. During the falling process, a transient rise of the diffraction intensity can be observed. This phenomenon is believed to result from perturbation and reorientation of the liquid crystal director after the voltage is removed. The experimental results indicate that the electrically controlled diffraction in the liquid crystal grating is reversible.” has been added at lines 131-136 in the revised manuscript.
Figure 3. Diffraction efficiency of the first-order diffraction as a function of time (a) when the device is operated with an applied voltage of 5.0 V, and (b) when the applied voltage is switched off.
Line 136: After recalculating the diffraction efficiencies, the modulation of the refractive indices (Δn) should also be recalculated. Furthermore, the values of Δn should be given with not more than 2 significant figures (e.g. 0.0019 instead of 0.001917). Giving values with 4 significant figures requires much higher precision of measuring the diffraction efficiency and LC cell thickness.
Response:
We really appreciate for the reviewer’s valuable comments. The values of Δn with 4 significant figures shown in Table 1 have been revised as values with 2 significant values.
Lines 148 – 155: Author contributions and Funding sections are not filled in completely.
Response:
The author contributions and funding sections have been added as:
“Author Contributions: conceptualization, Chie-Tong Kuo and Shuan-Yu Huang; investigation, Chuen-Lin Tien and Rong-Ji Lin; resources, Chie-Tong Kuo.; data curation, Rong-Ji Lin, Chi-Chung Kang, and Bing-Yau Huang; formal analysis, Rong-Ji Lin and Chi-Chung Kang.; writing—original draft preparation, Rong-Ji Lin; writing—review and editing, Chie-Tong Kuo.
Funding: This research was funded by the Ministry of Science and Technology (MOST) of Taiwan under Grant MOST 106-2221-E-035 -072 -MY2 ,the FCU/CSMU Project No. 17I42401 and FCU/CSMU 106-002.” at lines 181-186 of the revised manuscript.

Round 2
Reviewer 2 Report
The authors have answered in detail the questions and remarks of my colleague reviewer and me, and they have made changes to the manuscript, which significantly improve its clarity and readability.
My opinion is that the revised paper can be published as it is.